# The Relationship between Estrogen-Related Signaling and Human Papillomavirus Positive Cancers

**DOI:** 10.3390/pathogens9050403

**Published:** 2020-05-22

**Authors:** Claire D. James, Iain M. Morgan, Molly L. Bristol

**Affiliations:** 1School of Dentistry, Philips Institute for Oral Health Research, Virginia Commonwealth University (VCU), Richmond, VA 23298, USA; cdjames@vcu.edu; 2VCU Massey Cancer Center, Virginia Commonwealth University (VCU), Richmond, VA 23298, USA

**Keywords:** human papillomavirus, cervical cancer, head and neck cancer, oropharyngeal cancer, estrogen, estrogen receptor

## Abstract

High risk-human papillomaviruses (HPVs) are known carcinogens. Numerous reports have linked the steroid hormone estrogen, and the expression of estrogen receptors (ERs), to HPV-related cancers, although the exact nature of the interactions remains to be fully elucidated. Here we will focus on estrogen signaling and describe both pro and potentially anti-cancer effects of this hormone in HPV-positive cancers. This review will summarize: (1) cell culture-related evidence, (2) animal model evidence, and (3) clinical evidence demonstrating an interaction between estrogen and HPV-positive cancers. This comprehensive review provides insights into the potential relationship between estrogen and HPV. We suggest that estrogen may provide a potential therapeutic for HPV-related cancers, however additional studies are necessary.

## 1. Introduction

Human papillomavirus (HPV) is the most common sexually transmitted infection in the United States. Furthermore, the virus accounts for approximately 5% of all worldwide cancers [1,2,3,4,5,6,7,8]. These cancer-causing types are designated ‘high-risk’; human papillomavirus type 16 (HPV16) is the most common high-risk genotype, linked to 50% of cervical cancers and around 90% of HPV-related head and neck squamous cell carcinomas (HPV+HNSCCs) [4,9,10]. Numerous studies over the years have related estrogen and the expression of its receptors (ERα, ERβ) to HPV infections and to HPV-associated cancers. There are two schools of thought regarding whether estrogen and ER expression increases the incidence and severity of HPV and its related cancers, or if estrogen and its receptors could be exploited therapeutically for the treatment of infections and HPV-related lesions. There is evidence to support either claim. This timely review seeks to summarize what is currently known in the field, and to suggest strategies to move forward.

## 2. The Viral Genome, Proteins, and Lifecycle

Human papillomaviruses (HPVs) are small, double stranded DNA viruses. While there are numerous types of HPV, this review will focus on alpha-papillomaviruses with the propensity to infect mucosal epithelial cells, and are further characterized as high-risk by their predisposition to immortalize human keratinocytes and cause pre- and malignant lesions. Among these high-risk alpha-papillomaviruses are HPV 16, 18, 31, 33, 45, 52, and 58. These viral strains are estimated to contribute to at least 90% of all HPV-related cancers and are all included in the newest vaccine, Gardasil 9 [11,12,13]. While this prophylactic vaccine is instrumental in preventing future infections, there are currently no HPV-specific antiviral drugs to treat existing HPV infections and HPV+ cancers.

All HPVs share common genetic structures that consist of approximately 8000 base pairs, encoding around eight open-reading frames (ORFs) [14,15,16]. These ORFs are transcribed from a single DNA strand and can be divided into three functional parts: a largely non-coding region referred to as the long control region (LCR), the early (E) region, and the late (L) region [16]. The LCR contains *cis* elements that are necessary for the control of viral replication and transcription [17,18,19]. The early region encodes proteins (E1, E2, E4–E7) that are transcribed from an early promoter, and are responsible for the replication and transcription of the viral DNA, as well as structural regulation of the virus, and the principal regulators of viral oncogenesis [16,20]. The late region proteins (L1 and L2), are transcribed from the late promoter, and are responsible for the structural components that comprise the non-enveloped icosahedral capsid around the viral genome during the generation of progeny virions [13,15,21]. L1 is the major capsid protein and current vaccines are based on introducing virus-like particles (VLPs) of this protein to induce immunity towards the virus. In general, genotyping of the virus in the clinic is performed via PCR-based screens that probe for L1 consensus sequences [22,23].

The brunt of human cellular changes induced by the virus occur through interactions with the early proteins. HPV E2 is a DNA-binding protein that acts to recognize the viral origin of replication contained within the LCR. When E2 binds the origin, it recruits E1, the viral helicase and the only enzyme encoded for in the viral genome. E1 and E2 work together to recruit various cellular factors including host polymerases to allow for the replication of the viral genome [24,25,26,27,28,29,30,31,32,33,34,35,36]. E2 is also the main regulator of viral transcription. E4 acts late in the viral life cycle and is detected only in differentiated tissue [37]. E5 can interact with various cellular cytoskeletal components to facilitate viral assembly [38]. E5 may function both early and late in the viral lifecycle, and also serves as a minor viral oncoprotein [38,39]. Finally, E6 and E7 are the major viral oncoproteins that bind p53 and pRb, respectively, and target these cellular tumor suppressors for degradation [40,41,42,43,44]. These oncoproteins assist in maintaining viral episomes and stimulate cells to re-enter S-phase. A prolonged S-phase facilitates viral reproduction, as cellular polymerases are more freely available for the virus to exploit.

The HPV lifecycle is inextricably linked to the differentiation of infected epithelia [35,45]. Initiation occurs through infection of basal epithelia through micro-abrasions [13,46]. These basal cells provide the cellular proliferation component necessary for the initiation of viral replication. Early genes are expressed at a low copy number in the initially infected cell in a viral phase termed ‘initiation’. Under normal cell division, the daughter cell that loses contact with the basement membrane and begins migration through the differentiation process, withdraws from the cell cycle [47]. Conversely, HPV-positive epithelia continue to undergo the cell cycle and support DNA synthesis, even in the upper layers of the stratified epithelia [48]. However, the differentiation component is not completely halted. In the upper layers of HPV-positive epithelia, viral amplification occurs, increasing the viral load to high-copy numbers (1000+ copies per cell) [48]. Eventual terminal differentiation of these cells initiates the expression of L1 and L2 and the formation of the viral capsid around each copy of the viral DNA [49]. As with normal epithelia, upper layers are eventually shed. Shedding of these upper layers allows for the release of mature virions to repeat the viral infection process in a new host.

Circumventing the normal loss of cellular division over time is thought to be how HPV replication leads to tumorigenesis [50]. Loss of p53 and pRb allows for infected cells to replicate, and also allows for the accumulation of mutations in the cellular genome [6,7,51,52,53]. While productive viral infections rely on the expression of HPV in its episomal form, integration events can occur; these events are known to confer cellular growth advantages, and are thought to be a hallmark of cervical cancer progression [54,55,56]. Integration events are less well characterized in other HPV- cancers such as HPV+HNSCC. In cervical cancer, the integration of the HPV genome into the host leads to persistent expression of the viral oncogenes E6 and E7 [53]. Moreover, E2 is known to transcriptionally regulate and repress E6 and E7; in many integration events, the expression of E2 is interrupted, thus altering E2s ability to repress these viral oncogenes [53]. In both cervix and HNSCC, integration events do not seem be specific in regards to where they integrate into the host DNA. It has been shown that the integration at fragile sites throughout the human genome are more common than other locations. Moreover, the Myc locus is frequently associated with cervical cancer integration sites [54,57,58,59].

## 3. Estrogen, Estrogen Receptors, and Estrogen-Receptor Signaling

17β-estradiol, the predominant form of circulating estrogen, is a steroid hormone that plays a vital role in both sexes. It is well known that estrogen is exceptionally important in the regulation of the menstrual cycle in females and is highly abundant during a female’s reproductive years [60]. The significant decrease in estrogen synthesis, due to lowered function of the ovaries, causes women to enter menopause and is the reason women experience most menopausal symptoms [60]. Additionally, bone health, cardiovascular health, fertility, glucose homeostasis, immune function, neuronal function, and other homeostasis regulations, have all been linked to estrogen signaling [54,55,56,57,58,59,60,61,62,63,64,65,66,67]. Circulating estrogen can function in an endocrine, paracrine, or intracrine manner and this hormone signals through binding to its receptors: ERα, ERβ, or GPER1 [68,69].

Estrogen is primarily sourced in the ovaries, however, it can also be synthesized in the adrenal glands and adipose tissue of both males and females [69,70,71,72]. This hormone is synthesized from low-density lipoprotein (LDL)-cholesterol, which is processed by various enzymes to androstenedione, which can ultimately be converted to any of the steroid hormones, including estrogen [69,73,74]. Both natural estrogens, various synthetic estrogens, and estrogen mimetic agents are used in the clinic for a variety of reasons, including post-menopausal estrogen replacement therapy, infertility, sexual function, contraception, prostate cancer, and more [75,76,77,78,79,80,81]. Estrogen (natural or synthetic) can enter the plasma membrane and interact with the intracellular nuclear receptors ERα or ERβ, or interact with the membrane-associated protein-coupled estrogen receptor (GPER1) [82]. If unspecified in this review, ER means both ERα and ERβ, or that receptor type was unspecified in the noted studies. Estrogen-receptor complexes can bind directly or in-directly to specific DNA sequences known as estrogen response elements (EREs), and estrogen-mediated effects can be both genomic and non-genomic in nature [69]. While numerous EREs have been reported, it is estimated that 35% of human genes regulated by the estrogen-receptor complex, do not contain ERE-like sequences [69].

ERα is encoded for by the gene ESR1, and is found on chromosome 6 [69,83,84]. ERβ is encoded for by the gene ESR2, and is found on chromosome 14 [69,85,86]. Both of these nuclear receptors have full-length versions, and several shorter isoforms via alternate start codons and alternative splicing events [69,84,85]. The gene coding for GPER1 is found on chromosome 7 [69,87,88]. This receptor does not share similarities with the nuclear receptors and is instead a typical G-protein coupled receptor (GPCR) which has been linked to rapid responses to estrogen through activation of intercellular signaling cascades through secondary messengers [69,82]. ERα and ERβ can both interact with several GPCRs to initiate similar signaling cascades. While these receptors can work in concert, there is also evidence that they can have competitive interactions, and various isoforms that do not have transcriptional activity can repress full-length activation [69,89,90,91]. For example, estrogen activates AP-1-dependant transcription via ERα, however, ERβ inhibits this mechanism. Moreover, estrogen bound to ERα induces transcription when linked to Sp1 in GCrich regions, but not when it is bound to ERβ [69,92,93,94]. Overall, mechanisms involving estrogen-receptor complex signaling are vast, appear to be cell type specific, and can interact with various pathways including ras, Src and PI3 kinases, EGFR, IGF1, AP-1, STATs, ATF-2/c-Jun, Sp1, NF-κB, CREB, Elk-1, and many others [95,96,97,98,99,100,101,102,103,104,105,106,107,108,109,110].

## 4. HPV and Estrogen in Carcinogenesis

### 4.1. Cell Culture-Related Evidence for Estrogen Involvement in HPV Carcinogenesis

There is a solid foundation of evidence suggesting that estrogen signaling is involved in carcinogenesis. Estrogen-receptor complexes can regulate cell cycle progression and cell proliferation [111,112,113,114,115,116]. Historically, evidence has suggested that estrogen signaling in combination with HPV-related disease escalates disease progression [117,118]. Cell culture data, mostly conducted in breast cancer cell lines, has shown that estrogen treatment increases c-Myc and cyclin D1 expression and these effects are most notable at the G1-to-S transition, promoting entry into S-phase [119,120,121]. As previously mentioned, HPV relies on the maintenance of S-phase for its ability to replicate [2,13]. Early studies in HPV+ cell lines found that sex-hormone treatment can increase colony formation while having little effect on HPV− cell lines [99,122]. Similarly, early studies indicated that estrogen treatment could stimulate the production of HPV16 transcripts in SiHa cells [123]. Studies in the cervical cancer cell lines, CaSki and HeLa, determined that the ERα truncation, ER-α36, mediates estrogen-stimulated MAPK/ERK, and upregulation of this isoform increased the invasion, migration, and proliferation of these cell lines [124]. Finally, the 17β-hydroxysteroid dehydrogenase type 1 (HSD17B1), responsible for converting estrone to estrogen, has been found to be expressed at high levels in the HPV+ cervical cancer cells HeLa, SiHa and CaSki, suggesting that these cells are freely able to locally convert this circulating hormone [125]. Overall, the data indicates that estrogen promotes the oncogenic potential of HPV-positive cervical cell lines.

### 4.2. Animal Model Evidence for Estrogen Involvement in HPV Carcinogenesis

The Lambert laboratory and collaborators have demonstrated that the expression of the estrogen receptor α and E7 expression are intrinsically linked to the development and persistence of HPV-related cervical dysplasia and cervical cancer [117,126,127,128,129,130]. In this transgenic mouse model, estrogen acts as a co-carcinogen with E7 in the induction of cervical cancer. These studies have linked the expression of ERα in the cervix and supporting stroma, to estrogen-induced epithelial cell proliferation. They have also linked classical ERα pathway signaling to carcinogenesis in the cervix of K-14-E7 transgenic mice. Moreover, additional studies by the Lambert Laboratory have shown that E5 and E6 can also contribute to E7 and estrogen-induced cervical carcinogenesis, although the mechanism of this remains to be elucidated fully [131,132].

### 4.3. Clinical Evidence for Estrogen Involvement in HPV Carcinogenesis

Further evidence linking estrogen signaling to HPV-related carcinogenesis can be found in the clinic, although this data has confounding factors. The first reports linking sex hormones to genital cancers were noted in 1971. Since then, a number of additional studies have confirmed that the use of diethylstilbestrol (DES) by pregnant women, dramatically increased the risk of cervical cancer development in their daughters that were exposed in utero [122,133,134]. DES is a synthetic estrogen that was given to women at high risk for spontaneous miscarriage; use of this drug has been discontinued. However, the cervical cancer developed by the women exposed in utero is actually a rare form of adenocarcinoma of the vagina, not linked to HPV-related infections. Additional studies have linked the long-term use of hormonal contraceptives with increased risk of cervical dysplasia [122]. However, these studies do not take into account various confounding factors such as smoking, age, barrier contraceptive use, or other sexual practices [122,135,136]. Additionally, a previous analysis of HPV+ and HPV− genital lesions (cervical, vulvar, and penile) indicated that HPV-associated cervical lesions expressed high levels of hormone receptors, including ER [137]. The authors suggest that hormone signaling may act indirectly with HPV-infected epithelial cells, and can be implicated as co-factors in HPV-related cervical neoplasia [137].

## 5. HPV and Estrogen as a Possible Treatment Paradigm

### 5.1. Cell Culture-Related Evidence for Estrogen as an Hpv Treatment

There is significant evidence suggesting that estrogen may serve as a treatment modality in HPV-related lesions and cancers. The HPV18+ cervical cancer cell line, HeLa, is particularly sensitive to estrogen treatment [138,139,140]. Moreover, our laboratory confirmed HeLa estrogen sensitivity and further compared a number of HPV+ and HPV− epithelial cell lines for estrogen sensitivity; it was found that HPV expression confers specific sensitivity to estrogen, both through the expression of the viral LCR, and through expression of the viral oncogenes E6 and E7 [139]. This sensitivity to estrogen occurred in foreskin, cervical, tonsil, and HNSCC cell lines [139].

### 5.2. Animal Model Evidence for Estrogen as an HPV Treatment

A recent report demonstrated that high ERα expression in laryngeal squamous cell carcinoma (LSCC) correlated with improved survival [141]. While the studies were conducted with HPV− xenografts, it would be of interest to see if these studies showed a similar response in HPV+ xenografts. Xenografts of HeLa, SCC47, UMSCC104, and other HPV+ human cancer cell lines, as well as HPV+ patient-derived xenografts (PDXs), have been used for a multitude of studies analyzing other treatment modalities [142,143,144,145]. Thus, leaving room for additional avenues to research how estrogen and ER expression can affect HPV in other mouse models.

### 5.3. Clinical Evidence for Estrogen as an HPV Treatment

While there is limited evidence in animal models, patient studies provide evidence that the expression of the ER correlates with improved clinical outcomes, and that estrogen treatment could potentially enhance these outcomes. Recently, a number of studies have analyzed the expression of ERα in the TCGA [139,146,147]. These studies have all shown that higher ERα expression correlates with increased patient survival for HPV+ oropharyngeal cancer patients. Moreover, ERα expression in cervical cancer predicts favorable prognosis, and that loss of ERα enhances cervical cancer invasion and cancer progression [95,148]. Estrogen creams have been used vaginally for a number of years in postmenopausal women to alleviate many symptoms associated with menopause. Moreover, the low estrogen that women experience with menopause has been linked to problems with colposcopy reliability, as patients with hypoestrogen can mimic low-grade cervical changes [149,150]. Studies found that women, already presenting with low grade lesions, when treated with vaginal estrogen cream short-term, presented with more reliable colposcopy results [150]. Moreover, after long-term treatment, many patients showed negative colposcopy results in follow up exams [150]. Additionally, estrogen has been used as a treatment, in conjunction with surgery and other treatment modalities, of vaginal cancers both with and without HPV infections [151]. High rates of tumor regression and elimination were found in patients treated with intravaginal estrogen, both alone and in combination with other treatments [151]. Furthermore, survivors of many gynecological cancers often require estrogen supplementation. Studies have found that vulvar, vaginal, and cervical cancer survivors can use estrogen supplementation with no risk of disease recurrence [152]. Clomiphene, a non-steroidal estrogen analog utilized for fertility treatment, has also been shown that it may be able to treat HPV 16 and 18+ cervical lesions, as well as HPV 6+ and 11+ penile genital warts [153].

Further estrogenic clinical evidence can be observed in sex-related differences comparing both the frequency and the severity of HPV+HNSCC. Figure 1 presents a compilation of the CDC statistics from 2008–2012; it is approximated that the yearly total of HPV+ cancers for females are *n* = 23,000, whereas for males *n* = 15,800 [154,155]. Breaking down this data, the overwhelming majority of HPV+ cancers are oropharyngeal (HPV+OPC) in males (*n* = 12,600). By comparison, females only present approximately 3100 yearly cases of HPV+OPCs in this data set; suggesting a 4:1 ratio of men:women for HPV+OPC [154,155]. As of 2016, that ratio has increased to 5:1 [155]. Not only do men have a higher risk of developing HPV+OPC, men also have a higher risk of death from this disease [156]. Among these cancers, 80% were attributed to HPV types 16 and 18, and 12% were attributed to the 5 additional HPV types covered by the nonavalent Gardasil-9 [154]. This suggests that, over time, high vaccination rates should be able to almost eliminate these cancers. Unfortunately, as of 2017, only 49% of adolescents in the US are up to date on this HPV vaccine series [157]. While there could be numerous reasons why there are such discrepancies in the ratio of men:women affected by HPV+OPC, it is in an interesting premise that pre-menopausal women have much higher circulating estrogen levels [69]. While further studies are necessary, we propose that the high circulating estrogen in pre-menopausal women might assist in HPV-related clearance in the head and neck region of women. HPV-related cancers typically take years to develop; early assistance in clearing precursor lesions might be why post-menopausal women still exhibit lower levels of HPV+HNSCC while having similar circulating estrogen levels as men [158,159,160,161]. Men never have high levels, thus limiting estrogen’s advantages.

Reliable methods to test for HPV are difficult in the clinic due to low protein expression. For this reason, the surrogate marker, p16, is utilized to test for HPV infections in tissue sections. p16 shares an inverse relationship with the expression levels of pRb, which is degraded by E7 [13]. Although not a perfect marker because of false negative and false positive biopsy interpretations, it remains a routine way to screen for HPV infections in tissue biopsies [13]. Studies analyzing cervical lesions and cancers have found that ERα expression declined greater than 15-fold from normal tissue to cancer, and indicated a strong inverse correlation with increasing expression of p16 [162]. Moreover, breast cancer studies have indicated that p16 can be used as a prognostic indicator and predict how patients respond to hormonal therapy [163].

## 6. Estrogen, HPV, and Immune Function

Estrogen and related sex hormones are linked to immunity [61,164]. It has also been suggested, that due to higher estrogen levels, females have the enhanced ability to produce antibodies and mount more effective resistance to viral infections [164]. This might also lend to the HPV+HNSCC sex ratio discrepancies mentioned in the previous section. It is also interesting that a key function of HPV is to find ways to evade the immune system for persistent infection. HPV is well known to inactivate innate immune defenses, and the expression of E2, E5, E6, and E7 have all been shown to play roles in this downregulation of innate immunity [165,166,167]. Moreover, ER expression and signaling have also been linked to the regulation of the innate immune response [66]. Taken together, this demonstrates yet another way that estrogen signaling and HPV may be linked. It is likely that, as in Section 4 and Section 5, there is a complicated give and take relationship between HPV, estrogen, and the innate immune system. Moreover, ERα and ERβ have been shown to play both collaborative and antagonistic roles in relation to the innate immune system, adding an additional layer of complexity [168]. Given these complex interactions, further studies are necessary to elucidate whether HPV and estrogen may synergize or antagonize innate immunity.

## 7. Conclusions and Future Perspectives

Even with the prophylactic vaccine, HPV continues to present a worldwide disease burden. This is augmented by the lack of reliable tests for many of the areas infected by the virus, and by the lack of antiviral treatments. One possible antiviral-targeted approach could be related to estrogen signaling. There is clear evidence that estrogen-ER signaling has a multifaceted relationship with HPV-related infections and cancer. Figure 2A,B summarize what is currently known and addressed in this review. Conflicting evidence presents both a pro-carcinogenic and an anti-carcinogenic relationship in relation to estrogen and HPV. Cervical cancer cell culture studies indicate that estrogen can increase colony formation in some lines, and upregulated expression of a truncated version of ERα, ER-α36, increased the invasion, migration, and proliferation of some cell lines [99,122]. Whereas, other studies have shown, HeLa cells are particularly sensitive to estrogen, and estrogen specifically sensitizes numerous HPV+ cells when compared to HPV− cells, regardless of tissue of origin [138,139,140]. The overwhelming majority of animal studies in this area, have been conducted with the K-14-E7 transgene; these studies have made it abundantly clear that the expression of ERα is necessary for the formation of tumors in these models [117,126,127,128,129,130,131,132]. Few animal studies have looked at the estrogen and HPV relationship outside this tumor induction model. Animal studies with alternative models are necessary moving forward, and present a unique opportunity for additional research. Our lab will be expanding these animal studies in additional mouse models in the future. Finally, clinical data supporting estrogen’s role in inducing HPV-related cancers seems to be full of confounding factors, aside from confirming the K-14-E7 conclusions that the ER is present in these cancers [122,133,134,135,136,137]. Conversely, there have been a number of clinical studies indicating that the expression of the ERα correlates with patient survival, and that some HPV+ tumors have responded very well to estrogen treatment [95,146,147,148,149,150,151,152,153,154,155,156,157,162,163]. It is clear that there are sex-related differences of the number of instances and clinical responses of HPV+HNSCC [154,155,156]. This current data is observational, and there is a need to further develop this area, which presents a promising area of study.

Given the conflicting evidence, it is difficult to present overall conclusions. Extrapolating from current animal studies, it is clear that ERα does play a role in the development of cervical cancer. However, while it must be present for tumors to develop, it does not necessarily mean that estrogen causes cancer. Our previous cell culture work has indicated that HPV+ epithelia also express ERα; however, this allows for HPV-specific sensitivity to estrogen, both at the level of transcription via interactions with the LCR, and at the level of E6 and E7 expression [139]. If there are multiple mechanisms at the level of cell culture, there are likely to be even more mechanisms at play in both animal and clinical studies, and these should be investigated. It is our opinion that due to confounding factors on one side of the data, clinical studies favor estrogen as a useful treatment for HPV+cancers. These studies also need to be expanded, but present encouraging areas to develop. While outside the scope of this review, it is likely that other sex-related hormones, such as testosterone, may also play roles in the development and progression of HPV-related cancers; again, suggesting additional areas for promising future research. Largely, these findings highlight the complexity of the interactions and potential relationships between estrogen and HPV, and suggest numerous areas for further development.

## Figures and Tables

**Figure 1 pathogens-09-00403-f001:**
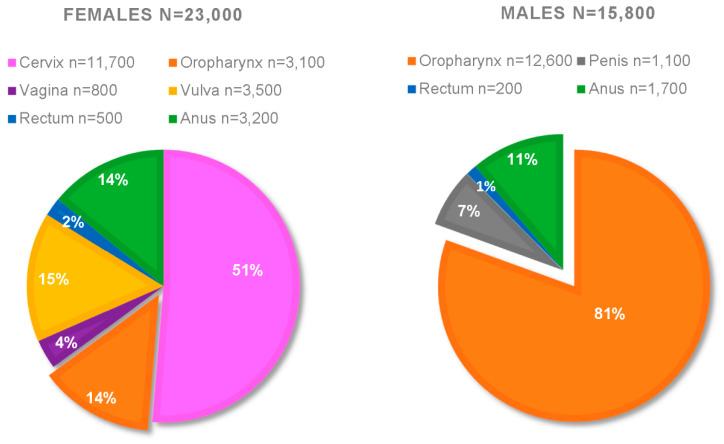
Average annual numbers of HPV-associated cancers by anatomical site and sex—United States, 2008–2012 from the CDC.

**Figure 2 pathogens-09-00403-f002:**
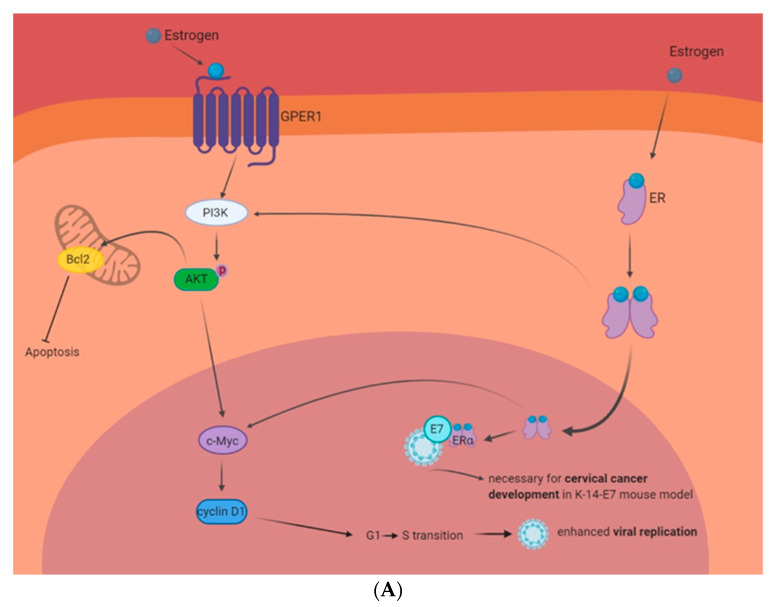
Estrogen-related signaling in HPV+ cells. Circulating estrogen interacts with its receptors and multiple interactions can contribute to the growth and death of HPV-infected cells. (**A**): HPV and estrogen in carcinogenesis. Interactions between estrogen and GPER1 or ER have shown to activate the PI3K pathway. This enhances Bcl2 expression and inhibits apoptosis. The PI3K pathway also activates c-Myc and cyclin D1. This enhances cell cycle progression and the promotion of the G1 to S transition enhances viral replication. Moreover, mouse studies have shown that the expression of HPV E7 and ERα are necessary for cervical cancer development. (**B**): HPV and estrogen as a possible treatment paradigm. Estrogen activation of phosphodiesterase 3A (PDE3A) stabilizes protein turnover of Schlafen 12 (SLFN12). SLFN12 binds to ribosomes and stops ER protein translation, including blocking Bcl2 and Mcl1. This induces cytochrome C release from the mitochondria and initiates apoptosis. Estrogen and ERα have also shown to interact with HPV 16 E6 and E7 to enhance HPV+ cell sensitivity to estrogen. Moreover, estrogen and ERα have been shown to interact with the HPV16 LCR to inhibit HPV transcription. Finally, HPV has been shown to enhance the expression of ERα and this correlates with enhanced clinical outcomes.

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
