# Peer review of "The Relationship between Estrogen-Related Signaling and Human Papillomavirus Positive Cancers"

_pathogens, 2020, doi:10.3390/pathogens9050403_

Round 1

Reviewer 1 Report

In this manuscript entitled “The relationship between estrogen-related signaling and human papillomavirus positive cancers”, authors provide insights into the relationship between estrogen and HPV driven carcinogenesis. They also present a synopsis of the studies that support estrogen as a useful treatment of HPV related cancers.

The review is well written and presents in a clear way both the evidence for the oncogenic role of estrogen signaling on pro-carcinogenic tissue and the evidence for the anti-carcinogenic role of estrogen signaling in HPV+ tumors.

Minor comments

1) Page 7: K-14-E7 mouse model should be replaced by K14E7 transgene

2) A schematic representation of ER signaling pathway and the interaction (positive/negative regulation) with other oncogenic or tumor suppression pathways would improve the manuscript

Author Response

Thank you so much for your suggestions.  

We think we understand which "K-14-E7 mouse model" reference you were referring to on page 7 and have changed it.

We appreciated the suggestion of a figure to outline HPV+estrogens effects and also agree that a schematic would improve this review.  We have added Figure 2A,B to address this suggestion and hope that it addresses this suggestion. 

Reviewer 2 Report

James et al. provide a timely and comprehensive summary of the conflicting data pertaining to the role of estrogen in HPV-related cancers. This review is well-organized and well-referenced, and it should provide a useful perspective for further investigations in this field. I only had a few minor points for the authors to consider.

1. None of the subsections were numbered except for the Introduction section.

2. In the section on the viral genome, proteins, and lifecycle the last paragraph talks about viral integration. There was no mention of the role that integration plays in disrupting regulation of E6 and E7 expression, and a brief mention of this important effect would be good background for the reader.

3. The section on "Estrogen, HPV, and immune function" should be expanded. There is only one sentence in this section about the relationship of estrogen to innate immunity, and the sentence doesn't indicate what effect estrogen has, just that it regulates innate immunity. More information about the role of estrogen in innate immunity and how this might synergize or conflict with HPV's effect would make this section more informative to the reader.

4. There seem to be frequent punctuation and grammatical errors, so thorough editing of this manuscript should be conducted before the final submission.

In reference #16 the title is in all caps.

Author Response

1- We apologize for any confusion in regard to the numbering of sections. We have now numbered all the sections and hope this addresses the reviewer's concern. 

2- We apologize that we did not give a more in-depth overview of integration in this review. Complete oversight on our part.  We have now added a couple of additional sentences to hopefully address this concern. 

3- Thank you for pointing this out. We acknowledge this area is complex and further development is necessary before we can postulate whether there is a defined synergistic or complementary role between HPV, estrogen, and the innate immune system.  We have expanded this section and now state this is also an area that needs further insight. 

4- This has been edited multiple times between individuals and we apologize for any punctuation and grammatical errors.  It has been edited again, however, without specific examples, we can only hope that we have addressed this concern to our best ability.

ref- Thanks for catching this. Our apologies, our reference software seems to want to capitalize this reference, we have changed it accordingly.